# Laser Beat-Wave Acceleration near Critical Density

**Ernesto Barraza-Valdez** [1,*], **Toshiki Tajima** [1], **Donna Strickland** [2] **and Dante E. Roa** [3]

[1] Department of Physics and Astronomy, University of California, Irvine, CA 92697, USA; ttajima@uci.edu
[2] Department of Physics and Astronomy, University of Waterloo, Waterloo, ON N2L 3G1, Canada; strickla@uwaterloo.ca
[3] Department of Radiation Oncology, University of California, Irvine, CA 92697, USA; droa@uci.edu
[*] Correspondence: ernestob@uci.edu

**Abstract:** We consider high-density laser wakefield acceleration (LWFA) in the nonrelativistic regime of the laser. In place of an ultrashort laser pulse, we can excite wakefields via the Laser Beat Wave (BW) that accesses this near-critical density regime. Here, we use 1D Particle-in-Cell (PIC) simulations to study BW acceleration using two co-propagating lasers in a near-critical density material. We show that BW acceleration near the critical density allows for acceleration of electrons to greater than keV energies at far smaller intensities, such as $10^{14}$ W/cm$^2$, through the low phase velocity dynamics of wakefields that are excited in this scheme. Near-critical density laser BW acceleration has many potential applications including high-dose radiation therapy.

**Keywords:** laser wakefield acceleration; beat wave; near-critical acceleration; fiber laser; endoscopic radiotherapy

## 1. Introduction

Laser wakefield acceleration (LWFA) allows us to make high energy acceleration of electrons [1–4]. Tajima and Dawson proposed using high-intensity pulsed laser (such as $10^{18}$ W/cm$^2$) to accelerate electrons with an accelerating gradient on the order of GeV/cm [1]. Their paper launched the laser wakefield acceleration (LWFA) branch of plasma physics, which was further aided by the advent of Chirped Pulse Amplification (CPA) and its advent also enabled LWFA realization [5]. The main allure and applications of LWFA has been to explore the energies that may not be covered easily by the conventional accelerator approaches, either in principle or by the ever-increasing cost and size of these. The energies of electrons accelerated by LWFA increase inversely proportional to the plasma density [1,3]. Thus, most of the explorations of LWFA so far have been in a density of plasma relatively far away from the critical density, the underdense regime (for a typical optical laser, the critical density is on the order of $10^{21}$/cm$^3$), so that the typical operating plasma density has been densities of $10^{17}$–$10^{19}$/cm$^3$. This is a gaseous plasma regime.

Despite LWFA having nearly half a century of history, there has yet to be sufficient exploration of laser-plasma acceleration near the critical density. Valenta et al. determined that electron densities of roughly $0.1n_c$ were necessary for high repetition rate, low energy, and short pulse lasers [6]. More recently, Nicks et al. further explored how one can achieve bulk acceleration of electrons by exploring the maximum energy achieved for different near-critical densities, laser intensities, and laser pulse widths [4,7]. We note that in the near-critical densities the gas plasma is replaced by other materials such as nanomaterials [8]. In such materials, by choosing the radius of nanotubes, for example, we can raise or reduce the average electron density that laser electromagnetic fields see. The outer shell electrons in such materials behave as if they are in a plasma state [9].

The purpose of this paper is to theoretically and computationally explore the near-critical density regime using laser beat-wave (BW) acceleration simulations. We will give a brief introduction to the theory of LWFA in the underdense, relativistic, high-intensity regime. Then we investigate the use of laser BW near the critical density with

low-intensity lasers, by using the well-benchmarked EPOCH 1D3V (1D and 3D in spatial and velocity calculations, respectively) and collisionless and relativistic Particle-in-Cell code with stationary ions [10]. We will show that in this near-critical density regime, electron energies up to 10 keV are obtained using intensities $\leq 10^{14}$ W/cm$^2$. These low laser intensities allow for the use of novel fiber technology described by Sha et al. [11], along with many applications such as for radiation therapy treatment, as described by Roa et al. [12].

This paper is structured in the following way: Section 2 will describe the theory of conventional underdense LWFA and beat-wave acceleration (BWA). Section 3 will describe BWA at near-critical densities and show results of PIC simulations with low-intensity lasers and near-critical density plasma targets. Section 4 will discuss applications to medicine. Section 5 will summarize and conclude this paper.

## 2. Underdense LWFA and BWA

LWFA in underdense plasma operates where the plasma density is much lower than the critical density $n_e \ll n_c$. Through the Stimulated Raman Scattering (SRS) process [13,14], an electromagnetic wave excites a plasma wave with $\omega_p$ following the frequency or energy conservation of:

$$\omega_p = \omega_0 - \omega_1 \tag{1}$$

where $\omega_p = \sqrt{4\pi n_e e^2 / m}$ and the '$p$' subscript denotes electron plasma, $n_e$ is the electron plasma density, '$e$' is the electron charge, and $m$ is the electron mass. Following the above frequency equation where $\omega_0$ is the incident wave and $\omega_1$ is the scattered wave, the wave number (or momentum) must also be conserved such that $\boldsymbol{k_p} = \boldsymbol{k_0} - \boldsymbol{k_1}$.

Similarly, Rosenbluth and Liu derived the process of exciting plasma waves using the beat wave of two lasers such that frequency difference was equal to the plasma frequency following Equation (1) [15,16]. This is beat wave (BW) excitation of plasma waves. When the two lasers are co-propagating, the Forward Raman Scattering (FRS) mechanism becomes significant and electrons can then be accelerated by the plasma waves [2,13,16–18]. We classify this as Beat Wave Acceleration (BWA). If laser fields for the pump $\boldsymbol{E_0}$ (higher frequency) and seed $\boldsymbol{E_1}$ (lower frequency) are near the relativistic range of $\propto m\omega_0 c / e$ then the ponderomotive force created by the beatwave of the two lasers drives an electrostatic longitudinal plasma wave $\boldsymbol{E_L}$ such that:

$$e\boldsymbol{E_L} = \nabla\left(\frac{e\boldsymbol{E_0} \cdot \boldsymbol{E_1}}{m\omega_0\omega_1}\right) = e\left(\frac{m\omega_p c}{e}\right)e^{i\,k_p x} \tag{2}$$

The phase velocity of this electrostatic wave (wake or plasmon) is driven by the group velocity of the lasers. For a plasma with density below the seed laser critical density ($n_{crit}$), the dispersion is given by $\omega = ck/\sqrt{1 - \omega_p^2/\omega^2}$ and the group velocity of the lasers is $v_g = \partial\omega/\partial k = c\sqrt{1 - n_e/n_{crit}}$ [19].

Therefore, in a highly underdense plasma the laser frequency $\omega_0, \omega_1 \gg \omega_p$ and both the laser phase velocity and plasmon group velocity reduce to $v_{ph} \approx v_g \approx c$. This is shown in Figure 1, where the 2D Fast Fourier Transform (FFT) of the transverse and longitudinal electric field of the laser beat wave PIC simulation in an underdense plasma ($n_e = 0.005 \times n_{crit}$) is plotted and shows the dispersion.

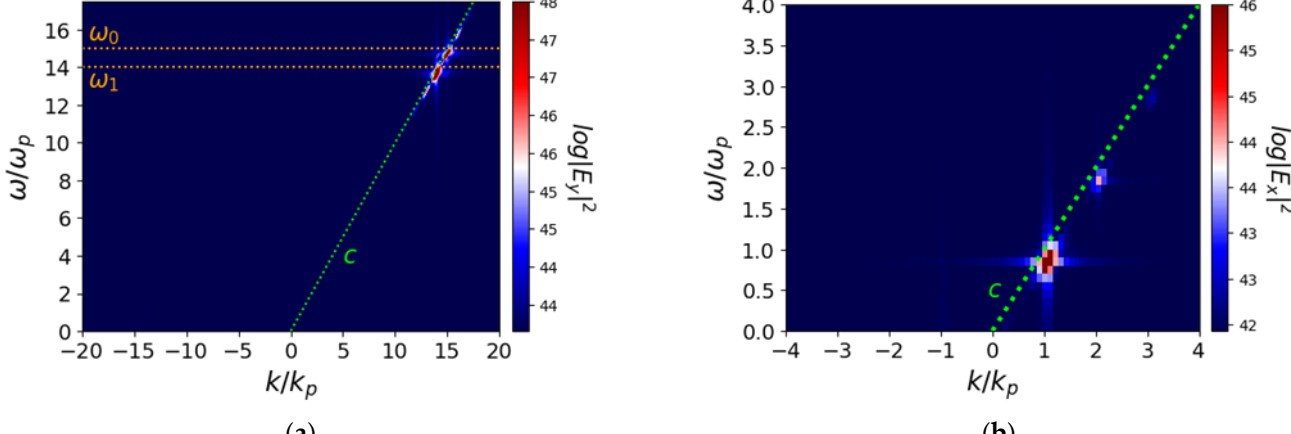

**Figure 1.** The 2D Fast Fourier Transform (FFT) of the (**a**) transverse ($E_y$) and (**b**) longitudinal ($E_x$) electric field in a 1D PIC simulation of highly underdense laser BW acceleration normalized to the plasma frequency ($\omega_p$) and plasma wave number ($k_p = \omega_p/c$). $\omega_0$ and $\omega_1$ are the two copropagating laser frequencies of the beatwave, indicated by the orange dotted lines in (**a**). The diverging color indicates the intensity of the fields in logscale. The speed of light dispersion (slope) is indicated by the lime green dotted line. This simulation used an equivalent of $a_0 \approx 0.5$ and $a_1 \approx 0.5$, beatwave lasers both with Gaussian shapes, wavelengths of 0.5 μm and 1 μm with 100 fs pulswidth and $n_e = 0.005 \times n_{crit}$.

A high phase velocity, much larger than the thermal velocity, allows the plasma wave to be stable and coherent against thermal plasma instabilities and allows the lasers to drive the plasma wave until saturation. In the relativistic regime, the wakefield amplitude saturates due to relativistic mass and detuning effects. The detuning time and length is defined as the length it takes for the laser beat wave and plasma wave to become more than $\pi/2$ out of phase such as described by Rosenbluth and Liu, Esarey et al. [3,15], and is typically very long. Thus, the saturated electrostatic wakefield is [2,15]:

$$E_L = \frac{m\omega_p c}{e}\left(\frac{16}{3}a_0 a_1\right)^{\frac{1}{3}} \tag{3}$$

where $a_{0,1} = \frac{e|E_{0,1}|}{m\omega_{0,1}c}$ is the normalized laser vector potentials for the pump, indicated by the 0 subscript, and the seed, the 1 subscript.

With the wake waves' high phase velocity, far away from the bulk thermal velocity of the plasma, the strength of these wakes will have to grow large enough, near the wave breaking limit [4,16], to be able to trap electrons from the fringes of the thermal distribution and accelerate them. If the amplitude of the wake waves is not strong enough, electrons will need to be externally injected at the high enough energies in order to be trapped. The range of velocities (energies) that the excited wake waves can trap and accelerate electrons is described by the trapping width velocity [20]:

$$v_{trap} = \sqrt{eE_L/mk} \tag{4}$$

where $E_L$ is the amplitude of the wake (longitudinal) wave and $k$ is its wave number. Thus, because $v_{ph} \gg v_{th}$ the wave number $k$ is small, making the trapping width small. Wakes cannot functionally trap and accelerate electrons from the bulk plasma thermal distribution. This again reinforces that the laser and wakes are stable from thermal plasma instabilities. Figure 2 shows the electric field and phase space of the highly underdense BW simulation in Figure 1. One can see that only at relativistic laser intensities ($a_0 \geq 1$) is the electrostatic wake amplitude driven large enough to trap a small population of electrons from the tail

end of the thermal distribution and accelerate them to relativistic energies. Additionally, Figure 2 shows that the observed wakefield amplitude (blue) matches Equation (3).

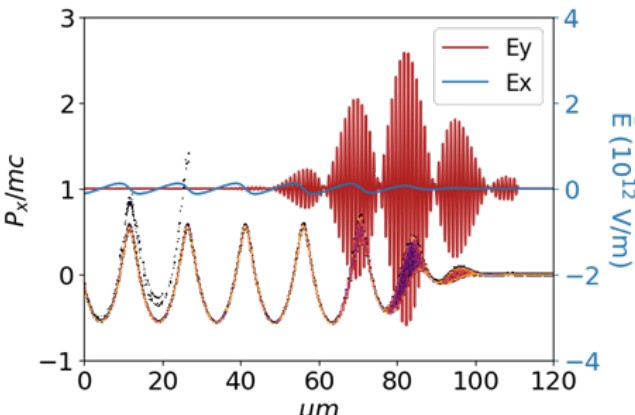

**Figure 2.** Electron phase space plot with electric fields of a 1D PIC simulation at 150 fs. The momentum is normalized to electron mass times the speed of light ($c$) on the left most $y$-axis ($P_x/mc$) vs. the position in the horizontal axis. The sequential color scale indicates the population density of electrons in that particular phase space point where bright orange indicates a large amount and purple or dark colors indicate small populations. The electric fields are indicated by the transverse (red) and longitudinal (blue) lines representing the laser and plasmon fields respectively, scaled to the right most $y$-axis (E $10^{12}$ V/m). This simulation used an equivalent of $a_0 \approx 0.5$ and $a_1 \approx 0.5$, beatwave lasers both with Gaussian shapes, wavelengths of 0.5 μm and 1 μm with 100 fs pulse width and $n_e = 0.005 \times n_{crit}$.

Once electrons with high enough velocities are trapped in the wakes, they can be accelerated. The maximum acceleration energy of trapped electrons is limited by the electron dephasing length and thus is proportional to $a_0$ and the Lorentz factor squared: $W_{max} = 2a_0\gamma^2 mc^2$ or $\propto a_0^2(\omega_0/\omega_p)^2$. For non-relativistic (nr), low intensities ($a_0 < 1$), in an underdense plasma we have no factor $2\gamma^2$ (due to the fact that relativistic dynamics of the phase velocity of the wake affects the extended dephasing length). If we assume in this case that the electron dephasing length is simply $\pi c/2\omega_p$, we obtain the low-intensity maximum electron energy which may lead to an expression such as $W_{max,nr} = \pi/2 \cdot mc^2 a_0^2(\omega_0/\omega_p)$. This type of energy would apply if the wakefield with nonrelativistic amplitude is excited but remains as a single wave. As we will see below, however, in the high-density regime of wakefields (near the critical density), we observe that a series of wakefields with different phase velocities tends to be excited.

As mentioned above, it is conventional wisdom that the maximum electron acceleration energy achieved using laser wakefield schemes is proportional to $(\omega_0/\omega_p)^2$ (due to the relativistic dynamics in underdense plasma). For this reason, we sought to see the limitations of the BWA with respect to density, similar to what was done by Nicks et al. [7]. We ran eight BWA simulations like those shown in Figure 2, with a pump and seed intensity given by $a_0, a_1 = 0.1$ and Gaussian pulse width of 100 fs, just below the non-relativistic regime. The kinetic energy distribution and maximum kinetic energies achieved after 1 ps are shown in Figure 3. Each simulation had uniform plasma density that is proportional to the critical density of the seed laser ($\lambda_1 = 1$ μm), as shown in the legend of Figure 3a.

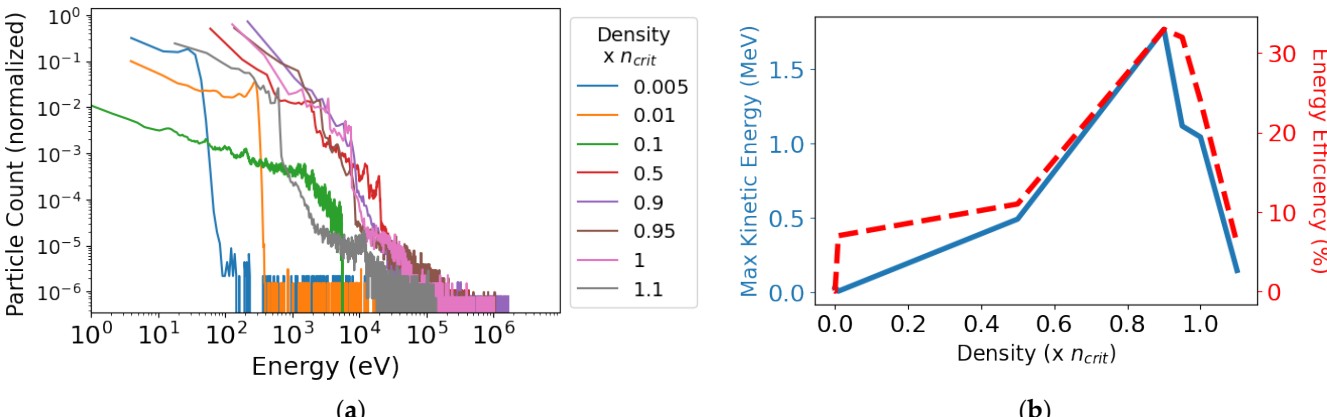

**Figure 3.** Energy distributions, maximum kinetic energies, and laser to total particle energy efficiency with respect to plasma density for BWA simulations after 1 ps using Gaussian lasers with intensities of $a_0, a_1 = 0.1$, and pulse width of 100 fs. The seed laser wavelength was held at $\lambda = 1$ μm while the pump wavelength was changed in order to satisfy Equation (1). (**a**) Normalized energy distributions (log-log scale) of eight different simulations with uniform plasma density from 0.005 $n_{crit}$ to 1.1 $n_{crit}$. (**b**) The maximum electron energies achieved after 1 ps with respect to plasma density (left *y*-axis, blue), and the efficiency of the total laser energy converted to total particle kinetic energy (right *y*-axis, red).

For low densities ($<0.1n_c$), Figure 3a shows that the excited plasma waves are barely able to trap and accelerate electrons from the fringes of the thermal distribution up to around 10 keV. However, at $0.1n_c$ we see that the plasma waves (with a low phase velocity at high densities which we will show in the next section) begin to push the thermal distribution out to higher energies. Above this density, a large tail of the distribution is formed and accelerated beyond 10 keV.

Figure 3b shows the maximum kinetic energy achieved after 1 ps (left *y*-axis, blue). Figure 3b also shows the energy efficiency or total particle kinetic energy of the plasma with respect to the total lasers' energy. As mentioned in this section, the low-density regime does not allow for the high phase velocity ($v_{ph} \approx c$) plasma waves to efficiently trap particles. Thus, external injection mechanisms would be needed for high energy acceleration. In contrast, near the critical density, particle trapping from the bulk thermal distribution becomes significant until around $0.9n_{crit}$. These particles can then be accelerated by the main wakefield to high energies. Additionally, one can see that the laser energy to total particle energy, efficiency, also increases with density which corresponds to the widening of the energy distributions in Figure 3a. Both the maximum kinetic energy and efficiency drops after $0.9n_c$, although it is unclear why. With this, the next section will focus on lower intensities in order to achieve electron energies of 10 keV for applications such as radiation therapy.

## 3. Near-Critical Density BWA

For low laser intensities ($a_0 < 0.1$) in highly underdense plasma, the plasmon wave amplitudes are not strong enough to trap electrons from the bulk thermal distribution and accelerate them to high energies as described above, in Figures 2 and 3a. However, when $n_e$ is approaches the critical density, additional physics must come into play. Nicks et al. showed electron energy scaling with respect to laser intensity ($1 > a_0 \geq 0.1$) for $0.1n_{crit}$ and $0.3n_{crit}$ in their Figure 5a,b [7]. Nicks et al. also showed that a somewhat discontinuous maximum energy gain of electrons as a function of $a_0$ (around 1) is observed. This arises from the transition between the nonrelativistic wakefields dynamics ($a_0 < 1$) and the relativistic one ($a_0 \geq 1$). Additionally, Figure 3 shows the electrons can be accelerated without external injection mechanisms near the critical densities.

Here, we show that using near-critical densities (of the seed laser) allows for low-intensity lasers to excite low phase velocity plasmons. This introduces the opportunity to pick up electrons in the bulk, which would not be possible in the conventional underdense operation of LWFA. This high-density, low-intensity BWA scheme makes it suitable for many applications.

For many industrial and medical applications, electron energies of greater than keV are required with low laser intensities of $\leq 10^{14}$ W/cm$^2$ (corresponding to $a_0 < 0.01$ for a 1 μm laser). For a 1 μm laser, the critical density is approximately $10^{21}$ cm$^3$, which is about one order of magnitude less than a conventional solid. This high density allows the BW accelerator to be done in atmospheric pressure rather than vacuum. Thus, our study will focus on using a 1 μm laser with plasma densities of approximately $n_e \approx 0.9 \times n_{crit} \approx 0.9 \cdot 10^{21}$ cm$^{-3}$. This density was chosen because it seems to provide the highest maximum acceleration energies along with maximum efficiency, as shown in Figure 3.

The excitation of low phase velocity plasmons with much lower laser intensities lends itself to the Enhanced Raman Forward Scattering (ERFS) method [21]. In the ERFS method, the laser intensities need not be equal as in the conventional BW case. Fisher and Tajima showed that ERFS allows for the lower frequency seed ($\omega_1$) laser's intensity to be 10% of the higher frequency pump ($\omega_0$) laser's intensity. In this case, the lower frequency laser seeds the growth of the plasma wave but there are also other seeding methods such as pulse shaping and electrostatic plasmon seeding. The smaller seed intensity further allows the usage of modern tabletop laser technology.

We now turn to the exploration of low-intensity, non-relativistic BW acceleration using the EFRS method. To represent a thin target of density $0.9 \times 10^{21}$ cm$^3$ in the simulation, the plasma is set to be between 1–20 μm thick (with ions left to be infinitely heavier and stationary). Figure 4 shows the transverse and longitudinal field spectrum for 20 μm-thick plasma. Figure 4a is similar to Figure 1a except that the seed frequency is now just above the plasma frequency ($\omega_1 \approx 1.1 \, \omega_p$). Additionally, the lasers propagate freely in vacuum before hitting the target. The free space propagation is clearly shown by the spectrum points that align with the speed of light in vacuum (lime green). In Figure 4b, one can see that there is excitation at $\omega_p$ with a broad range of wavenumber. This is due to nonlinear parametric processes and is allowed by the high-density plasma (near-critical) so that their phase velocities are low. The allowed group velocity of the photons (seed) is small ($\approx 0.3$ c) and so the ponderomotive force can excite plasma waves with small phase velocity which can trap and accelerate electrons from a cold thermal distribution with a temperature on the order of 10 eV. This allowed wide range of the phase velocities in nonrelativistic wakefields is the reason why we see this wide spectrum of plasma waves with frequency $\omega_p$. Additionally, in the relativistic regime the detuning length between the laser beat wave and plasma wave phases leads to saturation of the longitudinal plasma waves. However, from the resonant detuning length given by Esarey et al. [3], we see that for an $a_0 = 0.007$ and $a_1 = 0.004$, the detuning length is approximately 1 mm and much larger than our target length.

Figure 5 shows the phase space and electric field components for this simulation using a laser intensity of $2.5 \times 10^{14}$ W/cm$^2$ ($a_0 = 0.007$) with a pulse width of 2 ps. One can see that at 2 ps, the tail end of the laser pulses, and the longitudinal plasma wave amplitude (blue curve) has grown larger than the pump laser's amplitude (red curve). At 2.5 ps, the laser has left the simulation and one can see the strong plasma waves continue unimpeded and are able to accelerate electrons to keV energies.

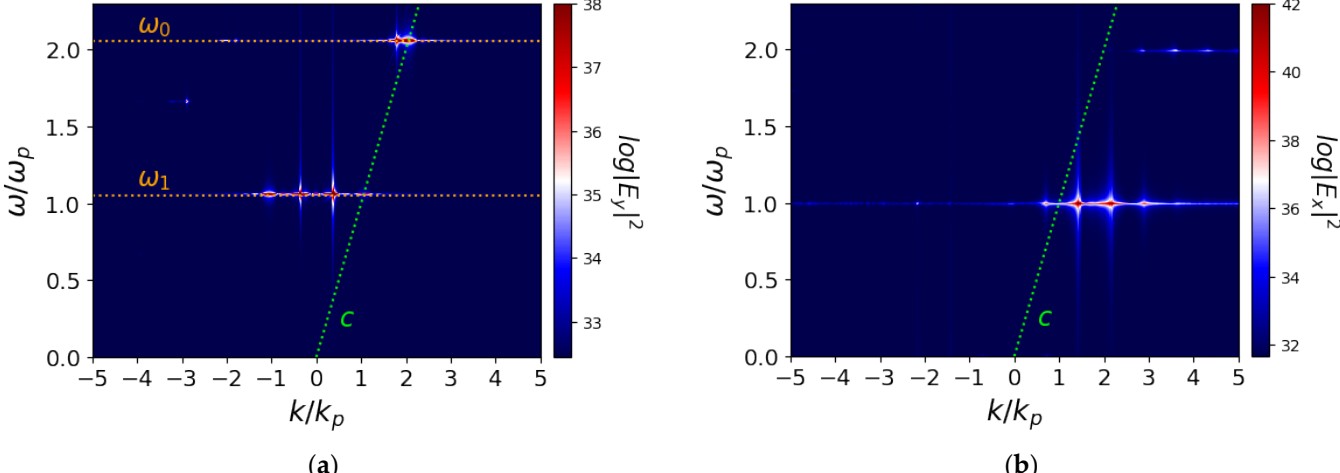

**Figure 4.** The 2D Fast Fourier Transform (FFT) of the (**a**) transverse ($E_y$) and (**b**) longitudinal ($E_x$) electric field in a 1D PIC simulation of near-critical density laser BW acceleration normalized to the plasma frequency ($\omega_p$) and plasma wave number ($k_p = \omega_p/c$). $\omega_0$ and $\omega_1$ are the two copropagating laser frequencies of the beatwave, indicated by the orange dotted lines in (**a**). The diverging color indicates the intensity of the fields in logscale. The speed of light dispersion (slope) is indicated by the lime green dotted line. The EFRS method is used with two Gaussian lasers with intensities of approximately $a_0 = 0.007$ ($2.5 \times 10^{14}$ W/cm$^2$) and $a_1 = 0.004$ ($9.7 \times 10^{13}$ W/cm$^2$), $n_e = 0.9 \times n_{crit}$, $\lambda_0 \approx 0.5$ µm and $\lambda_1 = 1.0$ µm, and the pulse width is 2 ps.

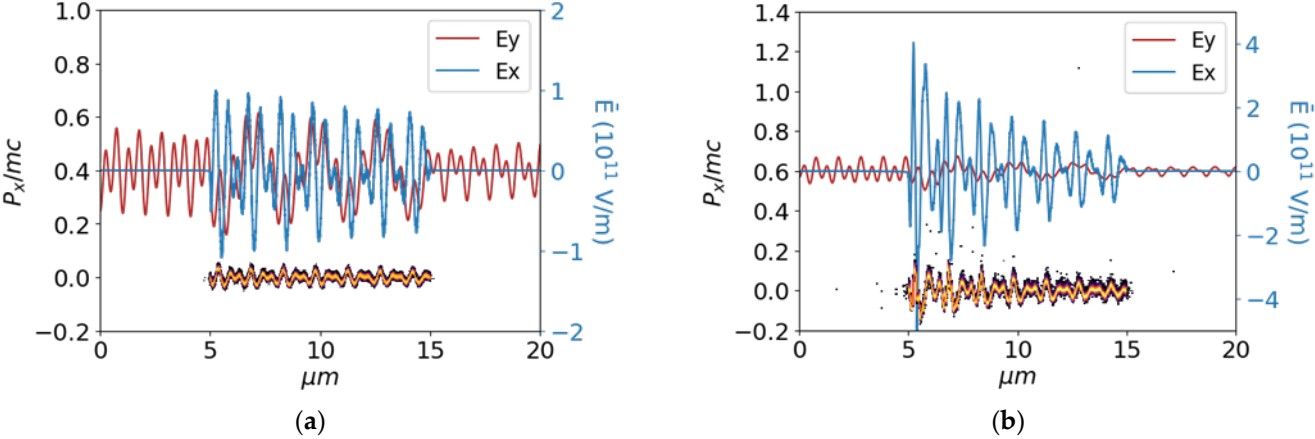

**Figure 5.** The phase space distribution and transverse and longitudinal electric fields for a 1D 10 µm plasma target at (**a**) 2 ps and (**b**) 2.5 ps. The momentum is normalized to electron mass times the speed of light ($c$) on the leftmost y-axis ($P_x$/mc) vs. the position in the horizontal axis. The sequential color scale indicates the population density of electrons in that particular phase space point where bright orange indicates a large amount and purple or dark colors indicate small populations. The EFRS method is used with $a_0 = 0.007$ ($2.5 \times 10^{14}$ W/cm$^2$) and $a_1 = 0.004$ ($9.7 \times 10^{13}$ W/cm$^2$) and $n_e = 0.9 \times n_{crit}$, $\lambda_0 \approx 0.5$ µm and $\lambda_1 = 1.0$ µm, and the pulse width is 2 ps.

Figure 6 shows the electron energies with respect to different simulations parameters. Figure 6a shows the electron energy distribution after 2.5 ps for varying thicknesses of the plasma target using laser intensities of $a_0 = 0.007$ ($2.5 \times 10^{14}$ W/cm$^2$) and $a_1 = 0.004$ ($9.7 \times 10^{13}$ W/cm$^2$). One can see that that below 5 µm, electrons are just barely accelerated above keV energies. This is most likely because the main plasma wave in the mix of wakefields is the one with a phase velocity near $c$ so that the main plasma wavelength corresponds to approximately 1 µm; this is confirmed by Figure 5. Thus, for small thicknesses there is not enough wake waves to accelerate electrons. However, there does not seem to be more enhancement in the electron energy spectrum as the plasma thickness is increased.

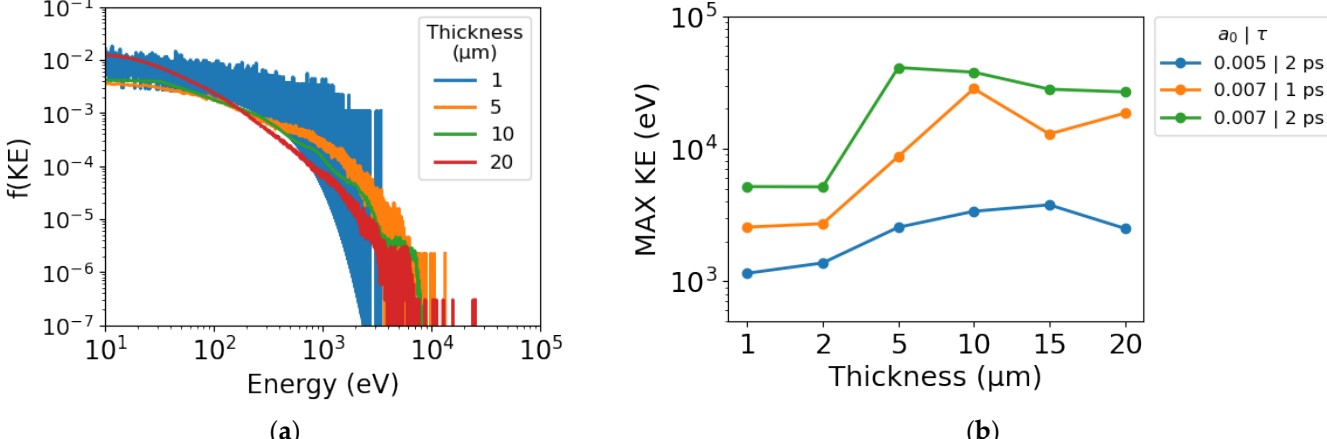

**Figure 6.** The electron energy gain with varying parameters: target length, pulse width, and intensity. (**a**) The energy distribution (log-log scale) of thin plasma target simulation after 2.5 ps with change of thickness. The intensity of the pump wave $2.5 \times 10^{14}$ W/cm$^{-2}$ and the seed intensity was one-tenth of that ($a_0 = \sqrt{10} \cdot a_1$). Pulsewidth of 2 ps. (**b**) Maximum kinetic energy (*y*-axis, log scale) of an electron found in simulation after 2.5 ps with respect to thickness (*x*-axis) given different laser parameters: (blue) $1.3 \times 10^{14}$ W/cm$^{-2}$ pump ($a_0 = 0.005$) with a pulse width of 2 ps, (yellow) $2.5 \times 10^{14}$ W/cm$^{-2}$ pump ($a_0 = 0.007$) with a pulse width of 1 ps, (green) $2.5 \times 10^{14}$ W/cm$^{-2}$ pump ($a_0 = 0.007$) with a pulse width of 2 ps.

In order to make this acceleration scheme more applicable, lower laser intensities than $2.5 \times 10^{14}$ W/cm$^2$ are chosen. Figure 6b shows the maximum kinetic energy found in the simulation after 2.5 ps for three different laser parameters: $1.3 \times 10^{14}$ W/cm$^2$ with 2 ps pulse width, $2.5 \times 10^{14}$ W/cm$^2$ with 1 ps, and $2.5 \times 10^{14}$ W/cm$^2$ with 2 ps pulse width. One can see that the intensity allows for higher electron energies, which is expected. In addition, increasing the pulse width to deliver more energy also increases the maximum kinetic energy that the electrons can achieve. Below $1.3 \times 10^{14}$ W/cm$^2$ ($a_0 < 0.005$), we saw that the plasma waves continued to grow beyond 2.5 ps.

## 4. Applications to Medicine

As we have found, the high-density operation (near the critical density) LWFA opens an avenue to make very compact electron acceleration (though electron energies are modest, such as 10's keV) with modest laser intensity. The near-critical density exceeds the usual gaseous plasma density without external injection mechanisms. The opportunity of using carbon nanotubes (and possibly other nanomaterials) to match this regime of density with changing its occupation ratio (such as the tube diameter) introduces an added flexibility and value of this regime operation [8,22]. Also, this brings in a flexible density ramping, if desired or necessary, for additional control in the acceleration process. The size of the target in this high-density LWFA further reduces the size of the accelerator from even the gaseous LWFA, which is already far smaller than the conventional accelerators. Additionally, near-critical density LWFA lends itself to low-intensity schemes such as EFRS. This allows the seed laser intensity to be 10% of the pump laser's and still achieve 10's keV electron energies.

Meanwhile, if and when we can use an electron accelerator so tiny that it can sit in front of the tumor of a patient (such as at the tip of an endoscope), the electrons for radiotherapy purposes need not have MeV of energies, as they do not have to penetrate a patient's body. In this case, the needed electron energies may be as low as 10's keV. The penetration depth of such electrons is short and reaches only the tissues facing the accelerator at the tip of a device such as an endoscope. As an example, the penetration depth of a 10 keV electron can be estimated using the modified relativistic Bethe formula for electrons in water and gives a mean penetration depth on the order of 10 μm [23]. Assuming a laser spot size of 10 μm, we estimate the dose of this electron beam per pulse is approximately $10^2$ Gy.

This study shows that our present regime of laser intensity and operation are within the reach of the fiber laser technology, see [11]. In addition, a new fiber grating technique [24] may help to create appropriate beat-wave resonance. Accordingly, this introduces a new possible way to operate an endoscopic electron radiotherapy using fiber laser. That is, an endoscopic radiation therapy. The surgeon would enter an internal part of a patient with an endoscope that has the HD LWFA attached at its tip. When he/she sees a tumor, he/she can turn on the HD LWFA at the suspected tumor. It could also be used to spray electrons after the surgical removal of a macroscopic tumor by endoscope, to make sure the remaining tissue can be devoid of active tumors (it may also be delivered as part of an acupuncture needle).

Such electrons may be used to address other therapy such as allowing for a hand-held radiation therapy device that can directly target superficial skin cancers such as melanoma [25–28]. Further, we could also employ a vector medicine (with high Z) that can guide itself toward the targeted tissue (such as cancer cells) that attracts and absorbs electrons preferentially to the vector molecules with its high Z distinction [29].

## 5. Conclusions

To summarize, we used the well-benchmarked 1D3V relativistic PIC code EPOCH and showed the dynamics of LWFA and BWA in underdense plasmas and near-critical density plasmas. In Section 2, conventional LWFA was discussed. Conventional LWFA relies on high laser intensities and underdense plasmas so that the group velocity of the laser ($\approx c$) can excite longitudinal plasma waves with similar phase velocity. Thus, a robust and coherent plasma wave train is excited with immunity from thermal plasma instabilities. It is conventional wisdom that lower plasma density allows for more stable wake waves and therefore larger electron energy accelerations [4]. However, we showed in Figure 3 that near-critical densities allow for low phase velocity plasma waves to trap electrons deep inside the thermal distribution and accelerate them to high energies.

In Section 3, we studied a low-intensity EFRS scheme to accelerate electrons to keV energies using low-intensity lasers (approximately $10^{15}$ W/cm$^2$) near the critical density. At the near-critical densities, we have shown that using laser BW with the ERFS method, nonlinear processes are excited. One of these processes is the growth of multiple small phase velocity plasma waves rather than the single large phase velocity wakefield, shown in Figure 5. This ensemble of plasmon waves with low phase velocities allows for efficient trapping of electrons from the bulk thermal distribution.

The EFRS scheme in a thin target has two advantages: the use of low-intensity lasers and microscopic acceleration length. The microscopic acceleration length is beneficial in many industrial and medical applications as discussed in Section 4.

**Author Contributions:** Conceptualization, T.T.; Data curation, E.B.-V.; Formal analysis, E.B.-V.; Funding acquisition, T.T.; Investigation, E.B.-V.; Methodology, E.B.-V.; Software, E.B.-V.; Supervision, T.T.; Validation, E.B.-V.; Visualization, E.B.-V.; Writing—original draft, E.B.-V. and T.T.; Writing—review & editing, T.T., D.S. and D.E.R. All authors have read and agreed to the published version of the manuscript.

**Funding:** The work was in part supported by the Rostoker Fund.

**Institutional Review Board Statement:** Not applicable.

**Informed Consent Statement:** Not applicable.

**Data Availability Statement:** The data presented in this study are available on request from the corresponding author.

**Acknowledgments:** We benefitted a great deal of advice and inspirations specifically on this or in general over the years from W. J. Sha, J. C. Chanteloup, G. Mourou, S. Iijima, and F. Tamanoi. For these we are very grateful.

**Conflicts of Interest:** The authors declare no conflict of interest.

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
