# Peer review of "Laser Beat-Wave Acceleration near Critical Density"

_photonics, doi:10.3390/photonics9070476_

Round 1

Reviewer 1 Report

The regime of BWA investigated by the authors is very interesting, original  and well presented. In my opinion it is quite poor the section of the potential applications, expecially because it will be confirmed by preliminary experimental results. The section 4 probably should be reduced and moved into the introduction. A new section 4 should be written about preliminary test needed to be done. The transition from the simulation of BWA regime to practical/medical applications is very long, so in my opinion it is mandatory to mention the next step regarding experimental campaign just to start testing the numerical simulations result.

Reviewer 2 Report

The manuscript “Laser Beat Wave Acceleration near Critical Density”, by Ernesto Barraza-Valdez et al., describes how a Beat Wave (BW) plasma accelerator operates when the plasma density is very close to the critical one. The excited wakes have a non-relativistic nature, although fields are still very intense, allowing to easily and efficiently trap background electrons and accelerating them, both backwards and forwards, up to energies in the range of few tens of keV. Moreover, since the plasma density is near critical, acceleration can be performed employing relatively low intensity lasers (O(10^14 W/cm^2)) and a seed laser less intense than the pump one (about a factor of 0.1). The Authors claim near critical density targets may be realized by use of nanotubes, whose outer shells electrons would behave like a plasma. Nanotubes radius can be varied for tuning plasma density and engineering targets. Finally, Authors speculate about how such a kind of tiny, low energy accelerator, may be realized by use of fiber optics lasers and built on an endoscope tip to deliver electron radio therapy directly on a cancer within a patience’s body.

The near critical density BW acceleration method proposed seems to be solid and already reported in literature so that there are little doubts it would work, at least by simulation results. The proposed application to radio-therapy is interesting and would clearly possess a neat advantage compared to methods currently in use. Still, the manuscript fails in reporting critical data necessary for assessing feasibility and efficacy of the proposed application, so that it cannot be accepted for publication in present form.

1) The missing critical information is the amount of extracted charge in the foreword direction: without this number it is impossible to estimate the amount of dose that could be deposed in cancerous tissues and if it would be enough to be effective. Extracted charge cannot be estimated from Fig. 5a, nor any of the reported plots/data.

2) Strongly related to point 1) is the fact that it is not clear to me, and the manuscript does not provide any hint, if the target would survive interaction with the lasers. This is not a minor issue, because fiber laser can operate at high repetition rates. If the single shot dose is not enough for effective radio treatment, repeated shots can build up the total dose to the desired level. Of curse, this needs the target to sustain the needed number of shots.

3) Authors should at least estimate the deposed dose, evaluate for which kind of cancers it is reasonable as a radio treatment and support their claims with some reference. In particular, the claim “In this case, the needed electron energies may be as low as 10’s keV”, line 241, strongly requires to be backed up by a reference.

4) English phrasing is sometimes a bit unclear. For example, sentence “Following the above ….   … such that k_p = k_0 - k_1”, lines 57-59, seems incomplete. “For non-relativistic …. … extended rephrasing length)”, lines 128-130, “The allowed group velocity … … velocity plasma waves”, lines 177-179, and “One can see … … pump laser’s amplitude”, lines 187-189, are rather involved and may be rephrased for sake of clarity.

5) To improve readability, I would suggest to add a legend to the plot in Fig. 5b.

Reviewer 3 Report

Manuscript entitled: “ Laser Beat Wave Acceleration near Critical Density ” by Ernesto Barraza-Valdez et al. reports an interesting theoretical/numerical investigation focussed on how to generate wakefields using Laser Beat Waves (BW). Authors perform 1-D Particle in Cell (PIC) simulations using the open code EPOCH to study the BW process. Simulations reports that using two counterpropagating laser pulses via BT process electrons are accelerated at energies up to keV in the nonrelativistic regime (I=10^14 W/cm^2).

The development of compact sources for particle acceleration via laser-plasma interaction and its respective biomedical and industrial applications represent nowadays a hot and interesting topic in laser-plasma community. The investigation carried out here can reduce the size of the particle source making it more flexible for potential radiation therapy applications.  

The manuscript merits publication. It is well written and clear. I recommend to the authors to include a “minor” comments that can help to the potential readers.

It is well known that one of the main problems of the BW process is the stability. The  phase shifts induced turns this process quite unstable. I would suggest to include a brief comment about it.

On Figure 5(a)-(b) the x-axis and y-axis are in logarithmic scale. I recommend to stress it in the label, or at least, in the caption of the figure. The same on the y-axis on Figure 5(b).

Reviewer 4 Report

Reviewer’s comments on ‘Laser Beat Wave Acceleration near Critical Density’ by Barraza-Valdez et al.

The authors present simulations using the PiC code EPOCH to simulate the interaction of lower intensity (1014 W cm-2) laser pulses with plasmas at densities near critical density.   Most current ‘wakefield’ acceleration schemes employ low density plasma and high intensity, ultra-short (fs) pulses lasers with the aim of creating a few high energy (e.g. MeV) electrons.   With the proposed acceleration, electrons are accelerated to several keV energies by picosecond duration pulses. It is suggested that electrons of keV energy are suitable for direct endoscopic or skin applications in medicine as the accelerating plasma could be incorporated in the tip of an endoscope or other device due to the lower laser intensities required.  The requirements for high vacuum enclosure of gas in which the plasma is created is also reduced.  The authors suggest that a large number of thermal electrons can be accelerated with their proposed acceleration technique.  The ideas are worth investigating and the simulation study is useful.   The presentation and use of language in the paper should be improved before publication.  I noticed the following issues which should be corrected or explained. 

  1. Line 84/85 Figure 1 caption. ‘normalized to the’ is repeated.
  2. Line 115/Figure 2 caption. I cannot understand the phrase ‘on the left most y-scale vs the position in the horizontal axis.’
  3. It is not clear what pulse shape variation in time is assumed in the simulations. Is the picosecond duration pulse varying as a Gaussian or ‘top hat’?  Some of the figures (e.g. figure 4) seem to suggest a pulse turning on to some constant value and then turning off after a few picoseconds (i.e. a ‘top hat’).   Does the temporal shape of the pulse affect the deduced electron energies?
  4. The Figure 3 caption looks to be incomplete. Not all parameters are specified.
  5. Line 156. The phase ‘ambient pressure’ is vague. Do the authors mean atmospheric pressure or something else? The laser irradiation of atmospheric pressure gas typically produces electron density of several 1019 cm-3, not values around 1021 cm-3  as implied in the text. Similarly the phrase ‘rather than vacuum’ is not helpful as in a vacuum, the electron density is zero.
  6. In a similar vein to point 3, the phrase line 230 ‘We no longer need the vacuum to hold gas plasma’ is meaningless.
  7. Is there evidence from the simulations if a large number of ‘thermal’ electrons are accelerated?  Can the authors quantify their statement in the conclusion (line 272): ‘This ensemble of plasmon waves with low phase velocities allows for efficient trapping of electrons from the bulk thermal distribution.’
